# Technology-Enhanced 4Active Intervention Impacting Psychological Well-Being and Physical Activity among Older Adults: A Pilot Study

**DOI:** 10.3390/ijerph19010556

**Published:** 2022-01-04

**Authors:** Weiyun Chen, Zhanjia Zhang, Bruno Giordani, Janet L. Larson

**Affiliations:** 1School of Kinesiology, University of Michigan, 830 N University Avenue, Ann Arbor, MI 48109, USA; 2Division of Physical Education, Peking University, 5 Yiheyuan Road, Haidian District, Beijing 100871, China; zhanjia.zhang@pku.edu.cn; 3School of Nursing, University of Michigan, 400 N Ingalls Street, Ann Arbor, MI 48109, USA; giordani@med.umich.edu (B.G.); janetlar@med.umich.edu (J.L.L.)

**Keywords:** geriatric population, physical activity, psychological well-being, wearable technology

## Abstract

Background: To increase psychological well-being and physical activity (PA) behaviors, our pilot study used the social ecological model as the framework to design the 4Active intervention, focusing on multicomponent exercise group lessons at the interpersonal level and self exercise enhanced by activity trackers at the individual level. The purpose of this pilot study was to examine the effectiveness of the two-level 4Active intervention in improving psychological well-being and PA participation in older adults living in retirement communities. Methods: Participants were 27 older adults with a mean age of 85.9 ± 9.3 years. Based on the two-arm, quasi-experimental study design, fourteen older adults (2 men, 12 women) living in one retirement community (RC) were assigned into the intervention group receiving the two-level 4Active intervention, whereas 13 older adults (1 man, 12 women) living in another RC were allocated to the active control group receiving group exercise intervention alone for eight weeks. One week before and after the interventions, the participants were pre-tested and post-tested in psychological well-being (i.e., life satisfaction, subjective happiness, positive affect, and negative affect) and weekly PA minutes (i.e., weekly walking, vigorous, moderate, and total PA minutes). The data were analyzed be means of descriptive statistics, independent sample t-tests, and ANCOVA repeated measures. Results: The results of ANCOVA repeated measures indicated that both groups maintained their slightly high or very high levels of life satisfaction, happiness, and positive affect over times. However, the two-level 4Active intervention group showed significant decreases in negative affect (F = 4.78, p = 0.04, η2 = 0.23) and significance increases in weekly moderate PA (F = 10.355, p = 0.004, η2 = 0.310) compared with the active control group over time. Conclusion: It is concluded that engaging in the two-level 4Active intervention including group-based multicomponent exercises and technology-enhanced self-exercises is more effective in decreasing negative affect and increasing weekly moderate PA METS-min in physically and cognitively frail older adults over time, compared with attending the group exercises alone.

## 1. Background

Promotion of psychological well-being is one of key factors contributing to successful aging. In multi-facets of psychological well-being, the hedonic facet refers to the extent to which an individual feels happiness, positive affect, and satisfaction with life [1,2,3]. Studies have suggested that high levels of psychological well-being are significantly associated with lower risk of frailty and lower future mortality and morbidity in geriatric population [2,4]. Likewise, evidence shows that regular physical activity (PA) is essential to maintaining quality of life and psychological well-being in older adults [5,6,7]. Therefore, promoting psychological well-being and regular PA in older adults aged 65 and older has been recognized as the most important public health priority [1,2,3,5,6,7].

A growing body of literature has explored the effectiveness of physical activity intervention in promoting psychological well-being and physical activity for older adults [7,8,9,10]. For example, Windle et al. [10] conducted a systematic review of 13 studies designed for sedentary older adults to participate in group exercise interventions taught by well-trained instructors. Four of these intervention studies, comparing tai chi or balance training conditions with a control group, found overall positive exercise effects on improving psychological well-being [11,12,13]. Moreover, studies reported no significant difference in improvement of psychological well-being between a tai chi intervention and a low-impact strength training exercise intervention [13,14]. Furthermore, multicomponent group exercise interventions, including balance, strength, and stretching, combined with a home training component showed a significantly greater improvement in psychological well-being among frail older adults as compared with home training alone [10].

Higher levels of psychological well-being are linked to increased PA [8]. The social ecological model posits that an individual’s PA behaviors are shaped by the dynamic interaction among individual, interpersonal, and community factors, as well as public policy. The likelihood of changing and maintaining PA behaviors will be increased when influential factors at the multiple nested levels of the social ecological model are addressed concurrently [7,15,16,17]. The individual level encompasses one’s knowledge, skills, fitness, attitudes, and personal characteristics that determine changes in their PA behaviors. The interpersonal level involves social interaction, social relationship, and social support, which are facilitating factors for an individual’s PA behavior change. Supporting the social ecological model, empirical studies have suggested that, in order to change PA behaviors of older adults, intervention strategies should focus on these changing factors that influence PA behaviors across multiple levels [7,16,17].

Up to date, a few multi-level PA intervention studies have focused on engaging older adults in walking with goal setting (individual-level), in a group (interpersonal-level), and on an improved walking path (community-level) [7,16]. While walking is an appropriate form of PA for older adults, the 2018 Physical Activity Guidelines recommend that older adults should engage in multicomponent physical activity, including aerobic endurance, muscle strength, flexibility, and balance exercises in order to ensure healthy aging [6]. Systematic reviews demonstrate that walking, resistance training, balance, and flexibility are the most common types (forms) of PA for older adults [15,18]. Further, systematic reviews suggest that multicomponent PA interventions that combine balance, strength, walking, and flexibility exercises were effective approaches to promote older adults’ initiation and maintenance of PA behaviors. More importantly, systematic reviews consistently conclude that older adults can gain physical and psychosocial health benefits from participating in multicomponent PA activities such as walking, strength, balance, and mind–body exercises [15,18].

Recent PA intervention studies have used wearable technology as a self-monitoring tool for promoting PA participation in older adults [19,20]. In a systematic review of activity monitors and PA, Lubans et al. indicated that 12 out of 14 studies using activity monitors significantly increased PA levels in terms of steps, distances, durations, and intensities [19]. Among wearable activity monitors, the Fitbit tracker is preferably used by individuals to self-measure their PA behaviors because it is an easy-to-use, reliable, and valid self-monitoring PA tool. Through pairing the Fitbit tracker with a smart phone and/or a computer via the Fitbit app, the individual can immediately check their real time steps taken, distance traveled, stairs climbed, and heart rate during exercises [20]. A recent study used the Fitbit tracker as a self-motivational tool to promote PA participation for older adults aged 60 years and older [20]. The results showed that all 35 older adults wore the Fitbit tracker daily for 12 weeks and downloaded their PA data records once per week using the Fitbit account via a synced computer. The results indicated that using the Fitbit tracker as the self-monitoring tool for PA is feasible and acceptable for older adults.

Previous studies have shown the evidence-based PA intervention strategies for promoting PA among older adults. However, to the best of our knowledge, PA interventions that synergistically implement multicomponent exercise group lessons at the interpersonal level and integrate Fitbit tracker into self-exercises at the individual level remains largely unexplored. Further, whether the synergistic implementation of the two-level PA is effective in promoting PA behaviors and psychological well-being among older adults is untapped territory. Thus, this is a pragmatic need to address the gaps in the PA intervention studies in the geriatric population. To increase psychological well-being and PA behaviors, our pilot study used the social ecological model as the framework to design the 4Active intervention, focusing on multicomponent exercise group lessons at the interpersonal level and Fitbit-based self-exercises at the individual level. The purpose of this pilot study was to examine the effectiveness of the two-level 4Active intervention in improving psychological well-being and PA participation in older adults living in retirement communities. This study hypothesized that both the two-level 4Active group and the active control group would increase psychological well-being and physical activity. The intervention group would show greater improvement in psychological well-being and weekly PA minutes than the active control group.

## 2. Methods

### 2.1. Participants and Study Setting

Two local retirement communities housing at least 75 residents each with similar SES and racial/ethnic background were recruited to participate through sending an invitation letter and follow-up emails, as well as meeting administrators in person. After receiving the study approval from each of the community administrators, we recruited older adults (via flyers, recruitment presentations, and word-of-mouth) to participate using the eligibility criteria. They were as follows: (a) 65 years old or above; (b) ability to walk for 10 m without human assistance (with cane or walker use allowed); (c) ability to read and write in English; (d) ability to exercise safely as determined by a healthcare provider; (e) ability to complete the study questionnaires and tests; and (g) providing informed consent. Individuals were excluded if they scored ≤10 on the Montreal Cognitive Assessment (MoCA) [21] and completed the Timed Up and Go (TUG) test for >30 s with a walking aid if needed [22]. One candidate with a MoCA score of 10 was excluded from the study. Upon providing the signed consent form, the participants were 31 older adults with a mean age of 86 ± 9.3 years. The University Institutional Review Board-Health Sciences and Behavioral Sciences (IRB-HSBS) reviewed and approved the study protocols (HUM0015827).

### 2.2. Study Design

This study used a two-arm, quasi-experimental design with a convenience sample method. One retirement community was assigned to the two-level 4Active intervention group and the other to the active control group. After two women and one man dropped off the study because of health conditions, 14 participants in the 4Active group (2 man and 12 women) received 8-week individual- and interpersonal-level intervention components. After one woman dropped off the study, 13 participants (1 man and 12 women) in the active control group received an 8-week interpersonal-level intervention component alone. The outcome measures were assessed at baseline and at post-test.

### 2.3. 4Active Intervention

*The two-level 4Active intervention*. The intervention group attended the two-level 4Active intervention for 8 weeks. At the interpersonal level, the participants attended a 45 min multicomponent exercise group lessons taught by well-trained coaches twice per week on site. Lessons used a structured format, consisting of 10 min of warm-up (e.g., walking patterns and stretching exercises), 25 min of functional fitness and balance, and 10 min of modified mind–body exercises. At the individual level, each participant used eight different weekly exercise task sheets we provided as their self-exercise resources. Each contains 4–5 exercises including walking, strength, balance, and flexibility, described with key learning cues and pictures. They were encouraged to engage in these exercises for 10–15 min daily (5 days per week) during their free-living time. They wore the Fitbit Charge 3 Fitness Activity Tracker (accelerometer-based, fitbit company in USA) daily for 5 days per week to self-monitor their PA engagement for meeting their self-determined weekly PA goals. 

Prior to distributing the tracker to the intervention participants, a synced Fitbit account was created. We demonstrated and explained the protocols for wearing the Fitbit tracker daily and the methods for self-monitoring their real-time steps and distanced traveled, calories burned, and heart rates. Every Monday morning, we distributed the fully charged Fitbit tracker to each participant, along with providing their previous week’s real-time daily PA steps report to discuss whether they met their self-determined weekly PA goals. Every Friday late afternoon, the tracker was collected, again, from each participant. Then, each participant’s real time PA data were uploaded, using the Fitabase software, while charging the Fitbit tracker over the weekend. 

*The active control group*. The active control group only attended a 45 min multicomponent exercise group lesson twice per week for 8 weeks. A well-trained wellness coach taught the lessons using the structured lesson format described as above on site.

### 2.4. Outcome Measures

Outcome measures were assessed one week before (baseline) and after the 8-week interventions (post-test) at the RCs. Trained research staff administered the outcome questionnaires to each participant individually at a time at baseline and post-test. In addition, to minimize potential self-section bias, the research staff conducted baseline characteristics measures. The MoCA was used to assess general cognitive status. A MoCA score of 26 and above is considered to be normal, while a score of 25 or less is considered to represent some level of cognitive impairment [21]. The TUG was used to assess functional mobility [22]. An older adult who takes ≥12 s to complete the TUG is considered at risk of falling [22]. We collected the participant’s demographic information: age, gender, marital status, education, self-reported body weight and body height, and body mass index (kg/m^2^). 

*The Satisfaction with Life Scale (SWLS).* The five-item SWLS was designed to assess an individual’s global life satisfaction [23]. The participants self-rated their global life satisfaction using a seven-point rating scale (7 = strongly agree, 1 = strongly disagree). The composite scores ranged between 5 and 35, with scores of 31–35 indicating extremely satisfied and scores of 5–9 representing extremely dissatisfied. The SWLS showed sound psychometric properties in the geriatric population [23]. The alpha coefficient of this study was 0.88.

*Subjective Happiness Scale (SHS).* The four-item SHS was designed to assess a person’s global subjective happiness in the past week [24]. The participants responded to the four items of the SHS using a seven-point Likert rating scale. The SHS was a valid and reliable scale used in the geriatric population [24]. In this study, the SHS had an alpha value of 0.818.

*The Scale for Positive and Negative Experience (SPANE).* The 12-item SPANE scale consists of the SPANE-Positive (SPANE-P) scale and SPANE-Negative (SPANE-N) scale [25]. The participants responded to the six-item SPANE-P to self-rate their general positive feelings and the six-item SPANE-N to self-rate their general negative feelings in the past month with a five-point rating scale, ranging from 1 (very rarely or never) to 5 (very often or always). The composite score of the SPANE-P and of the SPANE-N ranges from 6 to 30. Both the SPANE-P and the SPANE-N had sound psychometric properties [25]. In this study, the alpha value of SPANE-P was 0.922 and that of SPANE-N was 0.876.

*The International Physical Activity Questionnaire-Short Form (IPAQ-S)* [26]. The participants responded to seven questions on the IPAQ-S to self-report their performing amount of vigorous and moderate physical activities, walking, and sitting on a daily basis in the past 7 days. According to the guidelines [26], a specific formula with assigned Metabolic Equivalent Task minutes per week (MET-min/week) values (i.e., 3.3, 4.0, and 8.0) was used to calculate the MET-min/week score for walking, moderate-intensity PA, and vigorous-intensity PA. A total PA MET-min/week is computed as the sum of the walking + moderate + vigorous MET-min/week scores [26]. A minimum of at least 600 MET-min/week for the total PA is indicative of “minimally active”. A minimum of at least 3000 MET-min/week for the total PA is in the category of “HEPA active”. Not meeting the criteria for the minimally active category is classified into the “inactive” category [26]. The IPAQ-S has proven to be a reliable and valid measure of physical activity [27].

### 2.5. Data Analysis

Descriptive statistics of the outcome variables and the baseline variables were computed. Skewness and kurtosis of all variables were calculated to check their normality. Independent samples *t*-tests were conducted to examine if there was a significant baseline mean score difference in each outcome variable and each baseline characteristic between the two groups. The attendance rate of the group exercise lessons was calculated through dividing the number of lessons attended by the number of total lessons. The average rate and the range of wearing Fitbit tracker were computed through dividing the number of days worn by the number of total required days.

To determine effect size of each PA variable at the baseline and post-test, Glass’s δ was calculate considering a substantially different standard deviation of the two testing scores on each PA variable. Glass δ = 0.2 indicates a small effect size; Glass δ = 0.5 implies a medium effect size; and Glass δ = 0.8 represents a large effect size [28]. ANCOVA repeated measures were conducted to examine if there were intervention effects on each outcome variable separately. Time (pre-test vs. post-test) was treated as a within-subject factor and treatment condition (the intervention group vs. the active control group) was treated as a between-subject factor. Age, MoCA test, and TUG test were covariates in all ANCOVA repeated measure analyses of each hedonic well-being variable and of each PA variable to control for baseline differences. In addition, if a hedonic well-being variable or a PA variable was significantly different between the two groups at baseline, the baseline value of this variable was also controlled for in its ANCOVA analysis. None of the ANCOVAs violated the assumption of sphericity. Significance level was set at *p* < 0.05. All analyses were performed using SPSS version 27.

## 3. Results

### 3.1. Baseline Characteristics

Table 1 presents the baseline characteristics and Table 2 presents the baseline characteristics by group. None of the variables violate normality. Although the mean ages of the two groups were almost identical, the MoCA mean score of the 4Active group was 21.50, while that of the active control group was 24.08. Both were lower than 25, indicative of having mild and moderate cognitive impairment. The 4Active intervention group had a relatively higher TUG mean score (M = 20.21 s) compared with the active control group (M = 14.13 s). Both were higher than 12 s, indicating a high risk of falling. The intervention group had far lower mean weekly total PA METs-min than the active control group. Both groups were <3000 METs-min/week for a total PA, indicating minimally active. The independent sample t-tests revealed no significant differences in the three variables between the two groups (*p* > 0.05).

With respect to psychological well-being, the intervention group had significantly lower mean scores in life satisfaction and positive affect than the active control group (*t* = −2.529, *p* = 0.02; *t* = −3.911, *p* = 0.001). Therefore, the baseline life satisfaction and positive affect were controlled for in the corresponding ANCOVA. Although the mean scores of subjective happiness and negative affect in the intervention group were lower than those of the active control group, the independent sample *t*-tests found no significant differences in the two measures between the two groups (*p* > 0.05).

### 3.2. Adherence Rates

Regarding the adherence to the interpersonal-level intervention, the intervention group attended 88.4% of group lessons and the active control group attended 85.0% of the group lessons. For the adherence to the individual-level intervention, the intervention participants wore the Fitbit tracker on 93.9% of the required days.

### 3.3. Intervention Effects on Psychological Well-Being

Table 3 shows the baseline and post-test scores of psychological well-being by group and Table 4 presents the results of the ANCOVAs repeated measures. As seen in Table 4, for life satisfaction, happiness, and positive affect, there was no significant main effect of time and group, and no significant interaction of time with group. In contrast, for negative affect, there was a significant main effect of group (*F* = 8.017, *p* = 0.012, *η^2^* = 0.334) and a significant interaction between time and group (*F* = 4.782, *p* = 0.044, *η^2^* = 0.230), but no significant main effect of time. The results indicated that the change in negative affect over time was significantly different between the two groups. The 4Active intervention group showed a decreasing trend over time, whereas the active control group showed an increasing trend over time, indicating a beneficial intervention effect on negative affect.

### 3.4. Intervention Effects of PA Variables

Table 5 presents the scores of each PA variable at the baseline and the post-test and effect sizes (Glass’s δ) and Table 6 presents the results of the ANCOVAs with repeated measures. As seen in Table 6, there was no significant main effect of time and group, and no significant interaction between time and group for weekly walking, vigorous PA, and total PA METs-min. However, as seen in Table 5, the active intervention group had moderately increased changes in weekly walking and total PA (Glass’s δ = 0. 64; Glass’s δ = 0.79) and small increased changes in weekly vigorous PA over time. In contrast, the active control group increased vigorous weekly PA with a moderate effect size (Glass’s δ = 0.59), and decreased walking and total PA over time. 

As presented in Table 6, regarding the weekly moderate PA, there was a significant main effect of time (*F* = 7.470, *p* = 0.012, *η^2^* = 0.245) and group (*F* = 3.046, *p* = 0.022, *η^2^* = 0.208), and a significant interaction between time and group (*F* = 10.355, *p* = 0.004, *η^2^* = 0.310). The results indicated the 4Active intervention group demonstrated a significantly greater increase in weekly moderate PA minutes than the active control group over time. 

## 4. Discussion

This study was central to examining the effectiveness of the two-level 4Active intervention in improving psychological well-being and PA participation in older adults living in retirement communities. In particular, the majority of the study participants were “oldest-old” adults [27] and had mild or moderate cognitive impairment with a high risk of falling. Regarding the effectiveness in improving psychological well-being, supporting research hypothesis 1, the two groups maintained their respective slightly high or very high levels of life satisfaction, happiness, and positive affect over time, although most were the “oldest-old” [27] with mild or moderate cognitive impairments and a high risk of falling. This study is unique in that it is the first study to examine whether the two-level 4Active intervention and the active control group have positively influenced the psychological well-being in oldest-old adults with mild or moderate cognitive impairments and high risk of falling. The results confirmed that regularly engaging in multicomponent exercise group lessons has played an instrumental role in preventing psychological well-being from declines in physical and cognitive frail oldest-old adults. Consistent with the present results, a study reported that different types of PA interventions including strength, functional, and aerobic exercises produced beneficial effects on psychological wellbeing in older adults [8]. Likewise, examining the effects of an aerobic brisk walking condition or stretching and toning condition on psychological well-being, a study found that both groups showed significant positive increases in psychological well-being in older adults [29]. Further, studies have supported that PA is a promising lifestyle factor contributing to maintaining positive levels of psychological well-being [8,9].

Regarding the PA variables, partially supporting research hypothesis 1, the 4Active intervention group showed moderate-to-larger increases in weekly walking, total PA, and moderate PA minutes, but very small increases in weekly vigorous PA minutes over time. In contrast, the active control group showed relatively moderate decreases in weekly walking and total PA and a small decrease in weekly moderate PA minutes, but had a moderate increase in vigorous PA over time. The results are promising that, even though the participants in the intervention group are frailer in cognitive and physical functions and more inactive than those in the active control group at baseline, the intervention participants increased their weekly walking and moderate PA minutes, contributing to the increased weekly total PA participation over time. The results also showed that participating in multicomponent group exercise lessons was conducive to increasing vigorous PA in the older adults who are relatively less frail in cognition and function fitness.

Partially confirming research hypothesis 2, the intervention group showed a significantly pronounced improvement in negative affect and weekly moderate PA over time, compared with the active control group. These unique results are very promising because, at baseline, the intervention group had higher cognitive impairments, higher risk of falling, and lower PA engagement than the active control group; moreover, the intervention group was lower than the active control group in life satisfaction, happiness, and positive affect at baseline. Despite these baseline differences, after regularly participating in the multicomponent group exercise lessons and using the Fitbit trackers to self-monitor their own self-exercises during their leisure times, the far more physically and cognitively frail participants significantly decreased their negative feelings and increased weekly moderate PA, compared with their counterparts who only attended the group exercise lessons. 

In line with a previous study where older adults used Fitbit trackers to self-monitor their PA behaviors [20], the intervention participants in the study consistently wore the Fitbit tracker daily to self-monitor their own real-time PA behaviors over the course of eight weeks. Further, the study indicated that engaging older adults in individual-level intervention strategies is a promising approach to concurrently improve positive affect and PA behaviors, especially moderate PA. The individual-level intervention strategies focused on inviting the older adults to take ownership for self-initiating their own exercise plan in terms of frequency, intensity, duration, and types of PA during their free-living time with the study provided Weekly Exercise Task Sheets and using the Fitbit tracker to self-monitor their real-time PA engagement. The results suggest that, when using the Fitbit tracker to self-monitor their self-initiated PA behaviors at the individual level, the participants actively self-regulated their own health-enhancing PA behaviors, self-motivated themselves to achieve their self-referenced weekly PA goals, and self-monitored their progress toward meeting their goals. The active processes of engaging in the goal-setting, self-regulation, and immediate feedback actions are conducive to helping them feel autonomy and competence in living a healthy lifestyle.

Further, the study indicates that, when regularly participating in multicomponent exercises group lessons, the participants had opportunities to interact with one another and to develop a sense of community and social support. The social aspects of the group exercise intervention are beneficial to reducing the participants’ negative feelings, such as feeling lonely, isolated, disconnected, or sad [8,29,30]. Simultaneously, it is worth noting the group exercise lessons helped equip the participants with the knowledge and skills about appropriate ways to perform various forms of exercises. Possessing such knowledge and skills is further helpful for reinforcing the participants’ sense of autonomy and confidence for being physically active and socially engaged [8,29,30]. Taken together, this study suggests that combining the individual-level with the interpersonal-level intervention strategies is a more effective approach to reduce negative affect and to increase moderate PA for the physically and cognitive frail older adults, compared with the interpersonal-level PA intervention alone. 

Although the study has its unique strengths, it is important to discuss limitations. The pilot study did not use the randomized controlled trial design for the group allocation. Instead, we took the participants’ voluntary choices such as attending the group exercises and the two-level 4Active intervention into the consideration for the study design. Accordingly, we used quasi-experimental design for allocation of the two conditions with a lack of a no-intervention control group. In addition, the pilot study had a small sample size for both conditions nested within the two local retirement communities. To help gain benefits of the pilot study among more oldest-old adults, future studies should expand the two-level 4Active intervention to more retirement communities. With more retirement communities on board, future studies may consider to use cluster randomized controlled trial or alternative stepped-wedged design to strengthen the quality of the methodology and to engage more participants in the study simultaneously. Moreover, future studies may extend the length of the two-level 4Active intervention from 8 weeks to 12 weeks or longer. This pilot study only examined the immediate effects of the interventions on psychological well-being and PA behaviors. To help participants obtain lasting intervention effects, future studies focusing on examining short- and long-term intervention effects on these outcomes should be warranted. Regarding the measure of PA variables, the pilot study was limited to using the IPAQ-short form only. To increase the validity of measures of PA, future studies should use a combination of a subjective measure of PA with the validated IPAQ or other PA questionnaire and an objective measure of PA using research-graded accelerometers such as ActiGraph activity monitor or ActiPal activity monitor.

## 5. Conclusions

Regular participation in multicomponent exercise group lessons is effective in maintaining life satisfaction, happiness, and positive affect among the participants. High adherence rates of the two-level 4Active intervention components are more effective to reducing negative affect and increasing weekly walking, moderate PA, and total weekly PA minutes than attending the group exercise lessons alone among the physically and cognitively frail older adults. This study suggests that engaging in the group-based multicomponent exercises and technology-enhanced self-exercises, while actively using self-efficacy strategies is beneficial to maintaining mentally healthy and physically active aging.

## Figures and Tables

**Table 1 ijerph-19-00556-t001:** Descriptive statistics of baseline characteristics of all participants.

	*n*	%	Mean	SD	Median	Range	Skewness	Kurtosis
Age (years)			86.44	7.81	87.00	31	−0.72	0.52
65–74	2	7.4						
75–84	8	29.6						
>85	17	63.0						
MoCA			22.84	4.40	24.00	17	−1.37	1.64
≥26	6	22.2						
<26	19	70.4						
missing	2	7.4						
TUG (s)			17.40	8.71	15.21	42.56	2.71	9.64
≥12	7	25.9						
<12	19	70.4						
missing	1	3.7						
PA (METs)			1205.69	1195.26	975.00	4758.00	1.52	2.13
<600	10	37.0						
600–3000	12	44.4						
3000	3	11.1						
LS			26.74	6.87	29	27	−1.09	0.68
Happiness			5.28	1.36	5.50	5.50	−0.96	0.66
Paffect			24.45	4.24	25.20	15	−0.90	0.23
Naffect			11.08	4.40	11.00	15	0.59	−0.63

Note: MoCA = Montreal Cognitive Assessment; TUG = Timed Up and Go Test; PA: physical activity; LS = life satisfaction; Paffect = positive affect; Naffect = negative affect.

**Table 2 ijerph-19-00556-t002:** Independent t-tests of baseline difference between the intervention and the comparison groups.

	Intervention (*n* = 14)		Comparison (*n* = 13)		t	*p*
	Mean	SD	Mean	SD		
Age	86.07	9.51	86.85	5.81	−0.257	0.799
MoCA	21.50	5.81	24.08	2.10	−1.452	0.169
TUG	20.21	10.72	14.13	3.87	1.860	0.075
PA	809.61	935.93	1571.31	1323.86	−1.648	0.113
LS	4.77	1.54	5.97	0.84	−2.529	0.020 *
Happiness	4.89	1.66	5.69	0.80	−1.607	0.124
Paffect	3.67	0.70	4.51	0.39	−3.911	0.001 **
Naffect	2.09	0.83	1.60	0.55	1.764	0.090

Note: * = *p* < 0.05; ** = *p* < 0.01; MoCA = Montreal Cognitive Assessment; TUG = Timed Up and Go Test; PA: physical activity; LS = life satisfaction; Paffect = positive affect; Naffect = negative affect.

**Table 3 ijerph-19-00556-t003:** Baseline and post-test scores of psychological well-being by group.

	Intervention		Comparison	
	Mean	SD	Mean	SD
Baseline Life Satisfaction	23.86	7.71	29.85	4.20
Post-Test Life Satisfaction	23.14	6.85	29.46	5.24
Baseline Happiness	4.89	1.66	5.69	0.8
Post-Test Happiness	5.45	1.40	5.96	0.78
Baseline Positive Affect	22.01	4.20	27.08	2.33
Post-test Positive Affect	22.57	3.96	27.00	2.42
Baseline Negative Affect	12.54	4.98	9.62	3.31
Post-Test Negative Affect	11.36	3.15	10.54	2.70

**Table 4 ijerph-19-00556-t004:** Results of ANCOCAs for each component of psychological well-being.

Effects	Life Satisfaction			Happiness			Positive Affect			Negative Affect		
	F	*p*	η^2^	F	*p*	η^2^	F	*p*	η^2^	F	*p*	η^2^
Time	0.090	0.769	0.006	0.044	0.837	0.003	0.630	0.440	0.040	0.357	0.558	0.022
Time x Group	1.202	0.290	0.074	2.136	0.163	0.118	0.399	0.537	0.026	8.017	0.012 *	0.334
Group	1.202	0.290	0.074	0.000	0.989	0.028	0.399	0.537	0.026	4.782	0.044 *	0.230

*: *p* < 0.05.

**Table 5 ijerph-19-00556-t005:** Baseline and post-test results of PA variables by group.

	Intervention			Comparison		
	Mean	SD	δ	Mean	SD	δ
Baseline Walking	350.56	358.53			830.08	1083.12
Post-Test Walking	581.31	512.04	0.64	434.08	543.43	0.37
Baseline Moderate PA	83.33	207.82		558.46	411.21	
Post-Test Moderate PA	536.67	267.76	2.18	521.54	271.37	0.09
Baseline Vigorous PA	324.92	647.40		182.77	449.38	
Post-Test Vigorous PA	368.92	661.35	0.07	446.77	603.79	0.59
Baseline Total PA	809.61	935.93		1571.31	1323.86	
Post-Test Total PA	1546.83	948.60	0.79	1402.38	890.01	0.13

**Table 6 ijerph-19-00556-t006:** Results of ANOVAs with repeated measures for each variable of PA.

Effects	Walking			Moderate			Vigorous			Total		
	F	*p*	η^2^	F	*p*	η^2^	F	*p*	η^2^	F	*p*	η^2^
Time	0.193	0.664	0.008	7.470	0.012	0.245	1.039	0.318	0.042	0.962	0.337	0.040
Time x Group	2.777	0.109	0.104	10.355	0.004	0.310	0.530	0.474	0.022	2.445	0.132	0.096
Group	0.748	0.396	0.030	3.046	0.022	0.208	0.031	0.863	0.001	1.053	0.316	0.044

## Data Availability

The data that support the findings of this study are available on request from the corresponding author (W.C.). The data are not publicly available because they contain information that could compromise the privacy of research participants.

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
