# Peer review of "Technology-Enhanced 4Active Intervention Impacting Psychological Well-Being and Physical Activity among Older Adults: A Pilot Study"

_ijerph, 2022, doi:10.3390/ijerph19010556_

Round 1

Reviewer 1 Report

Your pilot study 4Active Intervention from a multilevel SEM-perspective and fitbit tracker with self-regulating impact is interesting and builds nicely on the present status of research literature. It is a pity that you have used a quasi-experimental design where the housing building is the discriminating variable for experimental/control-group but in itself that is an aspect of the community factor of the SEM-perspective. The study population of oldest-old community dwelling persons with mild cognitive impairments and falling risks is indeed novel, but it is a real pity that the samples are very small, and that your post-test is only a week after the intervention program (so we don't know anything about the lasting impact of 4Active). This makes the results of this pilot study of only average or lower interest. However, you could elaborate more in the Discussion about the pilot character of this pilot study: what are your lessons for a large scale follow-up study, and how then would you address the different methodological and design issues (real-experimental design, sample sizes, longer follow-up post test with even a third measurement, ....). And please reflect/discuss more the selfregulating effects of technology and possible improvements of the fitbit tracker and possible alterntives.

Please check editing (for instance in Abstract ANCOVA in stead of ANOVCA).

Author Response

Dear Reviewer 1:

Thank you very much for reviewing our manuscript and providing us with thoughtful comments.  After carefully reading and re-reading your very thorough comments and thoughtful suggestions, we have revised this manuscript accordingly. Without your very thoughtful guidance, we could not revise this manuscript with high quality. Again, we are so thankful for your time and wonderful help. In order for you to know where and how we addressed the comments, we used track changes to revise the manuscript.  I will provide point-to-point response to your specific comments with yellow highlight as follows.

Reviewer 1 comments:

Your pilot study 4Active Intervention from a multilevel SEM-perspective and fitbit tracker with self-regulating impact is interesting and builds nicely on the present status of research literature. It is a pity that you have used a quasi-experimental design where the housing building is the discriminating variable for experimental/control-group but in itself that is an aspect of the community factor of the SEM-perspective. The study population of oldest-old community dwelling persons with mild cognitive impairments and falling risks is indeed novel, but it is a real pity that the samples are very small, and that your post-test is only a week after the intervention program (so we don't know anything about the lasting impact of 4Active). This makes the results of this pilot study of only average or lower interest.

However, you could elaborate more in the Discussion about the pilot character of this pilot study: what are your lessons for a large scale follow-up study, and how then would you address the different methodological and design issues (real-experimental design, sample sizes, longer follow-up post test with even a third measurement, ....).

We discussed these issues by adding the limitation section. Please see page 9-10, lines 416-436.

And please reflect/discuss more the self regulating effects of technology and possible improvements of the fitbit tracker and possible alterntives.

We have elaborated on this. Please see page 9, lines 387-396.

Please check editing (for instance in Abstract ANCOVA in stead of ANOVCA).

We changed it to ANCOVA. Please see page 1, line 29.

Reviewer 2 Report

Thank you for the opportunity to review this paper. In general, the paper is well-written, and the topic is timely and important. The impact of the paper is, in my opinion, small; namely the sample size is rather small and there are some other drawbacks. Therefore, I currently cannot recommend publication, but I am willing to reconsider it after a major revision. My comments and concerns are listed below.

Title:

  • Include: “a pilot study” at the end of the title to indicate study type

Abstract:

  • Line 16-17: FitBit is a commercial brand, right? Please, use something like “self-exercise enhanced by activity trackers”.

Introduction:

  • In lines 57-60, you explain that multicomponent interventions are proven effective. Could you make it more clear in lines 87-96, how does your intervention differ from others? I know from reading the paper, that there are two novelties (two-level exercise and activity-tracking); this could be stated more explicitly here to show the reader what is the novelty of the study.

Methods:

  • Line 135: Rewrite as “they wore an activity trackers (FitBit).’’ Also add the model, manufacturer, country. Also include what type of sensor is it – accelerometer based?
  • The interventions are described somewhat vaguely. Could you add more information regarding volume, sets, repetitions, intensity, specific exercises? Perhaps some examples of take-home exercise sheets could be added as supplementary files?
  • Line 151: Were tests performed immediately after the last session; how many days between the last session and the testing?
  • IPAQ: I am not convinced that this is an optimal method to collect PA levels in older adults. Even your reference [24] reported only modest validity; furthermore, there are other studies reporting issues with this questionnaire’s validity/reliability (see below)

10.1186/s12874-018-0642-3

10.2188/jea.je20110003

Results:

  • Please reformat Table 5
  • Line 281: The effect size measure should be explained in the Data analysis section, with cut-off values for interpretation and appropriate reference.
  • I realize this is a pilot study, but could you nevertheless perform some sort of post-hoc statistical power calculation? This would increase the rigor of the paper and help future researchers determine their sample sizes

Discussion:

  • Please add 1-2 sentences at the beginning, summarizing the aim and rationale of the study

Conclusions

  • I think it is prematurely to say that two-level intervention per se is more effective than traditional interventions. The two-level group performed exercise on site, and also at home, which increased total volume. Is it not expected that they would show better results? I think your study shows that this type of intervention (two-level + activity trackers) is feasible and can increase well-being, but it is not clear whether it is superior to on site exercise of equal volume.

Author Response

Dear Reviewer 2:

Thank you very much for reviewing our manuscript and providing us with thoughtful comments.  After carefully reading and re-reading your very thorough comments and thoughtful suggestions, we have revised this manuscript accordingly. Without your very thoughtful guidance, we could not revise this manuscript with high quality. Again, we are so thankful for your time and wonderful help. In order for you to know where and how we addressed the comments, we used track changes to revise the manuscript.  I am providing a point-to-point response to your specific comments with yellow highlight as follows.

Reviewer 2 Comments

Thank you for the opportunity to review this paper. In general, the paper is well-written, and the topic is timely and important. The impact of the paper is, in my opinion, small; namely the sample size is rather small and there are some other drawbacks. Therefore, I currently cannot recommend publication, but I am willing to reconsider it after a major revision. My comments and concerns are listed below. 

Title:

  • Include: “a pilot study” at the end of the title to indicate study type

Thanks. We added it to the title. Please see the title page.

Abstract:

  • Line 16-17: FitBit is a commercial brand, right? Please, use something like “self-exercise enhanced by activity trackers”.

Yes, we changed from “Fitbit-enhanced….” To “self-exercise enhanced by activity trackers.”

Introduction:

  • In lines 57-60, you explain that multicomponent interventions are proven effective. Could you make it more clear in lines 87-96, how does your intervention differ from others? I know from reading the paper, that there are two novelties (two-level exercise and activity-tracking); this could be stated more explicitly here to show the reader what is the novelty of the study.

 Thanks a lot, we added information to strength the novelty of the study. Please see page 2-3, lines 89-110.

Methods:

  • Line 135: Rewrite as “they wore an activity trackers (FitBit).’’ Also add the model, manufacturer, country. Also include what type of sensor is it – accelerometer based?

We wrote the model: Fitbit Charge 3 fitness activity tracker (accelerometer based, fitbit, USA). Please see page 4, line 158-159.

  • The interventions are described somewhat vaguely. Could you add more information regarding volume, sets, repetitions, intensity, specific exercises? Perhaps some examples of take-home exercise sheets could be added as supplementary files?

We described the information already, “At the individual-level, each participant used eight different Weekly Exercise Task Sheets we provided as their self-exercise resources. Each contains 4-5 exercises including walking, strength, balance, and flexibility, described with key learning cues and pictures. They were encouraged to engage in these exercises for 10-15 minutes daily (5 days per week) during their free-living time.”  Since this is self-practice during their free living time and try to engage them in self-responsibility with autonomous choices, we only can encourage them to do their preferred exercises and use the fitbit tracker to self-monitor their exercise time and steps.  

  • Line 151: Were tests performed immediately after the last session; how many days between the last session and the testing?

We stated one week after the last group exercise session. Please see page 4, line 173 with yellow highlight

  • IPAQ: I am not convinced that this is an optimal method to collect PA levels in older adults. Even your reference [24] reported only modest validity; furthermore, there are other studies reporting issues with this questionnaire’s validity/reliability (see below)

10.1186/s12874-018-0642-3

10.2188/jea.je20110003

 Thanks for the information, we will talk about it in the Limitation of the study. Please see page 10, lines 432-436.

Results:

  • Please reformat Table 5

I also noticed the messy format. This messy format may be related to the process of converting the table to the publication format. The editing team may help this matter.

  • Line 281: The effect size measure should be explained in the Data analysis section, with cut-off values for interpretation and appropriate reference.

We added the information in the Data Analysis section. Please see page 5, lines 224-227.

  • I realize this is a pilot study, but could you nevertheless perform some sort of post-hoc statistical power calculation? This would increase the rigor of the paper and help future researchers determine their sample sizes

Due to the pilot study, we do not think it is necessary to run a Power Analysis to determine the sample size.

Discussion:

  • Please add 1-2 sentences at the beginning, summarizing the aim and rationale of the study

Thanks. We added. Please see page 8, lines 342-346.

Conclusions

  • I think it is prematurely to say that two-level intervention per se is more effective than traditional interventions. The two-level group performed exercise on site, and also at home, which increased total volume. Is it not expected that they would show better results? I think your study shows that this type of intervention (two-level + activity trackers) is feasible and can increase well-being, but it is not clear whether it is superior to on site exercise of equal volume.

Our conclusion is to highlight the key findings of the results, we did not compare our two-level intervention with the traditional intervention. We only stated that the two-level intervention is more effective in improving affect and PA compared to the group exercise alone among the study participants, not extend it beyond the study scope.

Round 2

Reviewer 2 Report

The authors have addressed my concerns.